# How *Streptococcus mutans* Affects the Surface Topography and Electrochemical Behavior of Nanostructured Bulk Ti

**DOI:** 10.3390/biom12101515

**Published:** 2022-10-19

**Authors:** Agata Sotniczuk, Agnieszka Jastrzębska, Adrian Chlanda, Agnieszka Kwiatek, Halina Garbacz

**Affiliations:** 1Faculty of Materials Science and Engineering, Warsaw University of Technology, 02-507 Warsaw, Poland; 2Łukasiewicz Research Network—Institute of Microelectronics and Photonics, 01-919 Warsaw, Poland; 3Institute of Microbiology, Faculty of Biology, University of Warsaw, 02-096 Warsaw, Poland

**Keywords:** titanium, AFM, EIS, passive films, microbiological corrosion

## Abstract

The metabolization of carbohydrates by *Streptococcus mutans* leads to the formation of lactic acid in the oral cavity, which can consequently accelerate the degradation of dental implants fabricated from commercially available microcrystalline Ti. Microstructure influences surface topography and hence interaction between bacteria cells and Ti surfaces. This work offers the first description of the effect of *S. mutans* on the surface topography and properties of nanostructured bulk Ti, which is a promising candidate for modern narrow dental implants owing to its superior mechanical strength. It was found that *S. mutans* incubation resulted in the slight, unexpected decrease of surface nanoroughness, which was previously developed owing to privileged oxidation in areas of closely spaced boundaries. However, despite the changes in nanoscale surface topography, bacteria incubation did not reduce the high level of protection afforded by the oxide layer formed on the nanostructured Ti surface. The results highlight the need–hitherto ignored–to consider Ti microstructure when analyzing its behavior in the presence of carbohydrate-metabolizing bacteria.

## 1. Introduction

The global dental market is growing, and it is expected to exceed $13 billion by 2023 [1]. This is a consequence of aging demographics, which inevitably boost demand for dental implants. Currently, pure Ti is one of the most commonly used materials in commercial dental solutions proposed by leading implant manufacturers, such as Strauman^TM^ and Nobel Biocare^TM^ [2]. This popularity is due to its biocompatibility and high corrosion resistance combined with relatively low stiffness [3,4,5,6]. The success of dental procedures is governed not only by the biomaterial selected but also by the quality of bone tissue at the implantation site [7]. Standard dental implants of 3.7–4.1 mm in diameter create difficulties in patients with atrophic alveolar bone and require special bone augmentation treatments which enhance the risk of complications [8]. Moreover, bone augmentation increases treatment costs and prolongs the healing time after implantation [7,8]. Apart from the quality of bone tissue, difficulties with the installation of implants with a standard diameter can also occur in the case of patients with limited space in the oral cavity [9,10]. Insufficient distance between replacement and adjacent teeth increases the risk of bone resorption after implantation [11]. One response to these issues is to fabricate narrow dental implants (diameter < 3.5 mm) [9]. However, clinical studies have revealed that commercially available pure Ti is not a good option for narrow dental implants, owing to the possibility of fractures occurring over the long term [9,12]. This is associated with insufficient mechanical strength [13], which can be enhanced by grain refinement to the nanoscale using large plastic deformation techniques [14,15,16,17]. Nanostructuring allows not only the reduction of the diameter of the Ti implant, but it also allows its immediate loading, which has been confirmed by *in vivo* trials [10,18]. Nevertheless, long-lasting dental materials must simultaneously demonstrate advantageous mechanical and surface properties in the complex oral environment, which contains a variety of inorganic (e.g., fluoride ions [19,20,21]) and organic species (e.g., proteins [22,23] and bacteria [24,25,26,27]). Carbohydrate-metabolizing bacteria, such as *Streptococcus mutans (S. mutans)*, merit special attention. *S. mutans* produces lactic acid, thereby lowering pH, which was found to reduce Ti corrosion resistance [28]. Atomic force microscopy (AFM) examination revealed an increase in Ti surface roughness after immersion in artificial saliva enriched with lactic acid, which confirms the possibility of Ti degradation in an acidified environment created by *S. mutans* [29]. The following conclusions are valid for commercially used microcrystalline pure Ti and require verification for its nanostructured counterpart. Literature data show that Ti nanostructuring usually enhances corrosion resistance in simulated physiological solutions, which is associated with a high number of structural defects which, in turn, promote passivation [30,31,32,33,34]. However, the increased reactivity of nanocrystalline material can also lead to accelerated corrosion processes in more aggressive conditions [15,35]. Although nanostructured Ti for dental applications has been extensively studied in regard to fabrication methods [15,36], properties characterization [37,38,39], and commercialization [18], little is known about its behavior in the presence of oral bacteria [40,41]. Hence, this study seeks to shed light on the effect of *S. mutans* on the surface morphology and electrochemical properties of nanostructured bulk Ti. To the knowledge of the authors, this manuscript provides the first description of the effect of *S. mutans* on the surface morphology and properties of nanostructured bulk Ti and highlights the need—hitherto ignored—to consider the level of Ti defectiveness when analyzing surface–bacteria interactions.

## 2. Materials and Methods

### 2.1. Fabrication of Nanostructured Ti 

To refine Ti grains to a nanometric size, microcrystalline Ti (rod diameter = 50 mm) was subjected to multistage hydrostatic extrusion (HE) with an equivalent strain of 3.7. This large plastic deformation process reduced the Ti rod diameter to 8 mm, which is sufficient for the fabrication of dental replacements [3,42]. The methodology of HE was optimized at the Institute of High Pressure Physics of the Polish Academy of Sciences, and a detailed description of it is provided in [43]. For the purpose of studying adhesion of *S. mutans*, disks from nanostructured Ti (8 mm diameter, 1 mm thickness) were ground to #2400 and polished with a 0.04 μm silica suspension. Subsequently, the disks were ultrasonically cleaned in a 70% ethanol solution. 

### 2.2. Bacterial Growth and Adhesion to Nanostructured Ti Surface 

*S. mutans* (ATCC^®^ 25175TM, Thermo Scientific, Waltham, MA, USA) was provided by the Faculty of Biology at the University of Warsaw. Bacterial cells were cultivated on tryptic soy agar (TSA) for 48 h at 37 °C on glass Petri dishes. Then, *S. mutans* cells were harvested from the solid medium and resuspended in PBS with 10 g/L glucose [44] to obtain the required optical density OD600 = 0.627, as measured by a spectrometer (UV–Vis Evolution 210, Thermo Scientific, Waltham, MA, USA). To test for the presence of bacteria, 1 mL of the suspension was applied to a copper mesh, which was then sputtered with a gold layer (thickness 8 nm) and subjected to scanning electron microscopy (SEM) observations (HITACHI SEM/STEM 5500N, Texas Materials Institute, Austin, TX, USA). After successful microscopic evaluation, the nanostructured Ti disks were placed in 24-well plates and immersed in a bacteria suspension (2 mL for each sample). The samples were incubated at 37 °C for 24 h and 48 h. For each incubation period, six nanostructured Ti disks were subjected to immersion in the bacteria suspension, and one additional disk was placed in a pure PBS solution enriched with glucose (control group). The time of incubation was selected in light of studies related to microcrystalline Ti, which state that *S. mutans* biofilm growth stabilizes after 48 h of immersion in a bacteria-rich environment [28]. After the end of bacteria growth, the suspensions were transferred to a sterile 24-well plate to measure their pH values. Nanostructured Ti disks were then washed three times with PBS and immersed in a glutaraldehyde solution for 10 min. The fixed disks were washed with PBS and dehydrated with an ethanol solution. 

### 2.3. Surface Characterization 

A scanning electron microscope (HITACHI SEM SU-70) was used to evaluate the process of bacteria adhesion to the nanostructured Ti surface. Disks for SEM examination were sputtered with a gold layer (thickness 8 nm). The number of bacteria was calculated using the Micrometer program [45,46]. An atomic force microscope (Dimension Icon Bruker AFM) was used to provide a quantitative description of the effect of bacteria adhesion and its metabolization products on surface morphology. The AFM was connected to a NanoScope V controller. A silicon ACT series probe (App Nano) with a nominal spring constant of ca. 37 N/m and a nominal radius of ca. 6 nm was selected for topographical investigation of the surface of the Ti disks. To ensure examination conditions with minimal impact from the external environment, the microscope was encased in a chamber supplied by the microscope’s producer. Prior to imaging, the drive frequency of the scanning probe (ca. 260 kHz) was adjusted with the built-in auto-tune function. Topographical information was recorded using standard tapping mode (TM) in air, in ambient conditions (18 °C and relative air humidity of ca. 28%). Subsequently, the topographical images were used for roughness analysis via the freeware Gwyddion (ver. 2.56). AFM examination was performed initially on the nanostructured Ti disks with a mirror-like, polished surface, which were subsequently immersed in an *S. mutans* suspension. The evaluation was repeated after the biofilm was removed from the nanostructured Ti surface by ultrasonication in ethanol and then in ultrapure water. Additionally, AFM analysis was carried out on the material prior to large plastic deformation (microcrystalline Ti with a mirror-like, polished surface). This verified whether the surface nanotopography of Ti is governed, not only by the surface preparation methodology, but also by the microstructure of the material. Data related to the number of bacteria, pH, and surface topography were statistically analyzed using one-way analysis of variance (ANOVA) and post-hoc Tukey tests. Significant statistical differences were identified for *p* < 0.05. The analysis of the changes in the nanostructured Ti surface morphology induced by bacteria incubation was supplemented by electrochemical evaluation of passive-layer protectiveness. Electrochemical measurements (electrochemical impedance spectroscopy (EIS) and potentiodynamic polarization (PD)) were performed in an artificial saliva solution (using Fusayama’s recipe [19]) for nanostructured Ti in the polished state and after removal of the *S. mutans* biofilm. All of the electrochemical tests were carried out with a three-electrode system: working electrode–Ti sample (WE), counter electrode–platinum (CE) and reference electrode–saturated Ag/AgCl (RE), connected to a potentiostat (Autolab PGSTAT 302N). EIS tests were performed at open circuit potential (OCP) from 10,000 Hz to 0.05 Hz (amplitude ±10 mV). PD measurements were conducted from −0.2 V to 2.5 V vs. OCP (scan rate 0.002 V/s). EIS measurements were performed after OCP stabilization (1 h after immersion). PD curves were recorded immediately after the EIS tests (ca. 2 h after immersion). A detailed description of the tested samples is provided in Table 1. 

## 3. Results and Discussion

### 3.1. Microstructure and Surface Topography of Nanostructured Ti

Figure 1 shows the microscopic evaluation of the microstructure and surface topography of the nanostructured Ti fabricated by large plastic deformation. It can be noticed that the multiple-pass HE process resulted in the formation of well-defined nanometric grains with a high dislocation density in their interiors (Figure 1). Although the nano Ti surface was polished to a mirror-like state, AFM examination showed that surface roughness had developed at the nanoscale (Figure 1). This could be associated with the refinement of its microstructure (Figure 1) by introducing a high number of grain boundaries during the multistage HE process [47]. Grain boundaries demonstrate enhanced reactivity compared to grain interiors; hence, they could act as privileged oxidation sites. Consequently, the closely spaced grain boundaries in nano Ti could be responsible for its developed surface topography at the nanoscale. This hypothesis was confirmed by the less-developed surface nanotopography reported for the microcrystalline Ti (the initial material for the large plastic deformation, Figure 2), which demonstrates a markedly less defective microstructure. It is widely known that surface nanotopography can moderate the adhesion of pathogenic (bacteria) cells [48,49,50,51]. Hence, our results highlight the importance of carrying out a separate analysis of the attachment of *S. mutans* to nano Ti substrates.

### 3.2. Bacterial Adhesion to the Nanostructured Ti Surface

The microscopic characteristics of nano Ti immersed in the *S. mutans* suspension are shown in Figure 3. Only single bacteria cells were visible in the case of the shorter incubation time (24 h). The number of attached bacteria cells increased significantly when the incubation period was extended to 48 h (Figure 3 and Figure 4, Table 2). The significant growth in the number of adhered *S. mutans* cells observed after the 48 h incubation period corresponds to a slight decline in the pH value of the growth medium (from pH = 7.8 to pH = 5.8—Figure 4, Table 2). This is associated with the formation of lactic acid owing to the metabolization of carbohydrates by *S. mutans* during biofilm formation [28]. Lowering pH, induced by the presence of lactic acid, accelerates microcrystalline Ti degradation, as confirmed by surface analysis [29] and ion-release measurements [52]. Corrosion of microcrystalline Ti in lactic acid was observed in the pH range from 1.0 to 8.5, and only slight differences in the extent of degradation were found in solutions with different pH values [52]. Thus, it can be assumed that 48 h incubation of *S. mutans* could induce surface degradation of nano Ti, despite only a slight decrease in pH. Moreover, it must be added that the pH value was measured in the bacteria suspension, whereas the real conditions existing directly between the biofilm layer and the nano Ti surface could be much more acidic [28]. 

### 3.3. Effect of Bacterial Exposure on the Surface Topography of Nanostructured Ti

Changes in nano Ti surface topography were evaluated both qualitatively (SEM) and quantitatively (AFM). SEM evaluation did not reveal any signs of degradation, such as local corrosion damage or microscale heterogeneities (Figure 3). However, slight changes in the nanotopography after 48 h of bacterial incubation were reported by AFM. It was found that immersing nano Ti in *S. mutans* suspension resulted in the slight smoothening of its surface at the nanoscale (Figure 5, Table 3). This could be associated with the huge number of grain boundaries, which privilege Ti dissolution in more aggressive conditions, such as that caused by lactic acid. Preferred degradation in the area of grain boundaries can result from their enhanced reactivity compared to the grain interiors. Selective dissolution within the more reactive grain boundaries could be responsible for nano Ti smoothening.

### 3.4. Effect of Bacterial Exposure on the Electrochemical Response of Nanostructured Ti

Changes in nanoscale surface topography can induce differences in functional properties, such as corrosion behavior. Since corrosion performance is of particular importance for long-term dental applications, it was necessary to investigate the protectiveness of the nano Ti oxide layer after the biofilm was removed. Electrochemical measurements performed under steady-state conditions (EIS tests) did not reveal significant changes in nano Ti corrosion behavior induced by *S. mutans* incubation (Figure 6). The flattened course of the phase-angle portion of the Bode plots indicates that a tight oxide layer, which was homogenous in structure, formed on the tested samples. For this reason, an electrical circuit with a single time constant (Figure 6) was selected to analyze the EIS results. The electrical circuit contained two resistors, R_s_ and R_p_, which corresponded to solution- and passive-layer resistance, respectively, and one constant phase element (CPE). The parameters of the CPEs (Q_in_ and n) were exploited to calculate effective capacitance (C_eff_) according to the Hsu–Mansweld equation [53]. This approach to designating capacitance values from a CPE was demonstrated as accurate in the case of analyzing the protective properties of passive films [54,55]. For both states, calculated C_eff_ values were in the range of 10^−5^ F × cm^−2^ (Figure 6), similar to those reported for nano Ti tested in physiological saline [33]. Calculated values of passive-layer resistance (R_p_) exceeded 10^7^ Ω × cm^2^, which confirmed the high level of protection provided by passive layers formed on nano Ti in the polished state, as well as after bacteria removal. As compared to the literature data, the R_p_ values designated for nano Ti were about an order of magnitude higher than those reported for conventional microcrystalline Ti as tested in Fusayama’s artificial saliva [19,56]. This can be associated with following factors: (i) the higher protectiveness of the Ti oxide layer governed by nanostructure [33]; (ii) a differential surface preparation procedure and thereby surface morphology, (iii) a differential time range between surface preparation and electrochemical testing, which significantly affects the properties of the native passive layers [23]. 

Slight differences in corrosion behavior were detected during polarization measurements (PD tests—Figure 7). It was found that the corrosion potential (E_corr_) was ca. 250 mV higher for nano Ti subjected to the bacteria environment. Generally, higher E_corr_ values denote better corrosion resistance. However, in the case of metals in the passive state, it is possible that the faster passivation, and hence inhibition, of the oxygen reduction reaction might be responsible for more negative E_corr_ values being recorded [47,57]. Hence, we claim that the accelerated oxidation of the polished nano Ti after its immersion in artificial saliva is the reason for the lower E_corr_ value compared to the nano Ti surface which was previously immersed in the *S. mutans* suspension. For both states, current density in the passive state (i_pass_) remained stable during polarization up to ca. 1.4 V (Figure 7). After exceeding ca. 1.4 V, a slight increase in the i_pass_ value was observed for the nano Ti after *S. mutans* incubation. According to the literature, the i_pass_ increase could result from the thickening of the passive layer, which does not balance the potential difference across the surface [58]. This indicates that the oxide layer formed on nanostructured Ti after bacteria removal demonstrates a greater tendency to further growth during anodic polarization. However, in further studies, this conclusion needs to be supported by additional surface analysis using spectroscopic techniques. 

## 4. Future Outlook

This study found that titanium surface nanotopography can be affected not only by the surface preparation procedure, but also by the material’s microstructure, e.g., grain refinement to the nanoscale (Figure 2). Therefore, it is essential to provide a comparison of the effect of bacteria on surface morphology and the properties for dental Ti with different microstructures (e.g., grain sizes) with surfaces prepared using the same methodology. Moreover, owing to the long-term character of dental implants, it is necessary to replicate *S. mutans* incubation on nano Ti surfaces that were previously exposed to bacteria suspension. Considering the fact that the level and character of surface roughness at the nanoscale can moderate antibacterial properties [59], a smoother nano Ti surface after *S. mutans* incubation could behave differently after repeated bacteria exposure. Finally, future works must provide an overall comparison of materials proposed for use in narrow dental implants. A comprehensive analysis could allow the indication of products that offer the most promising combination of mechanical performance and functional properties, such as corrosion resistance, in the complex environment of the oral cavity.

## 5. Conclusions

This work offers the first description of the effect of carbohydrate-metabolizing bacteria (*S. mutans*) on the surface morphology and properties of bulk nanostructured Ti, which is a promising material for use in commercial narrow dental implants. The results can be summarized as follows:Significant coverage of nano Ti surface by *S. mutans* was observed after 48 h of incubation in a bacteria suspension enriched with glucose. Bacteria adhesion and the beginning of biofilm formation were combined with a slight decline in pH, which could be associated with lactic acid formation owing to glucose metabolization.*S. mutans* incubation resulted in an unexpected, slight reduction in nanoscale surface roughness. Preferential nano Ti dissolution in the closely spaced grain boundaries induced by the pH shift to more negative values could be the reason for the observed phenomenon.Differences in surface nanotopography induced by *S. mutans* incubation provided virtually no changes in the protectiveness of the passive layer formed on the nanostructured Ti surface in the artificial saliva. This confirms the high application potential of bulk nanostructured Ti in the area of dental implantology.

## Figures and Tables

**Figure 1 biomolecules-12-01515-f001:**
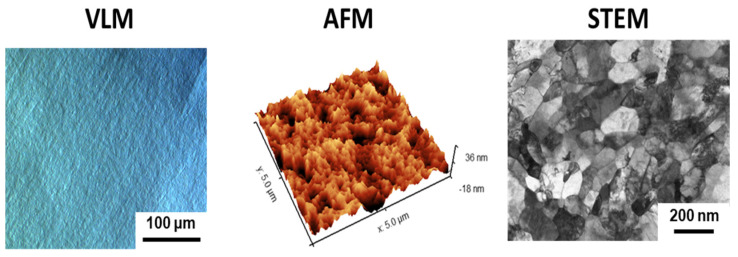
Evaluation of nanostructured Ti surface topography (VLM micrograph and AFM map) origins from its refined microstructure (STEM image).

**Figure 2 biomolecules-12-01515-f002:**
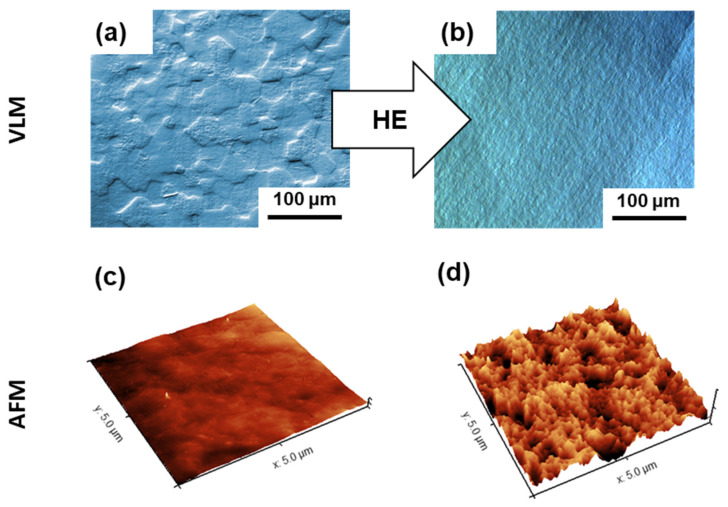
Microstructure of the (**a**) microcrystalline Ti and (**b**) nanostructured Ti (VLM micrographs with DIC contrast), (**c**) microcrystalline Ti, and (**d**) nanostructured Ti (AFM images).

**Figure 3 biomolecules-12-01515-f003:**
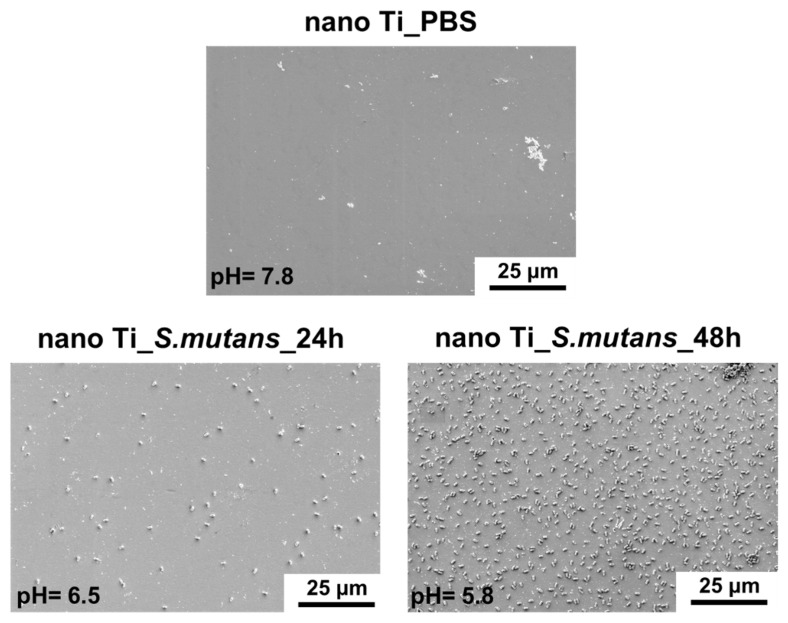
*Streptococcus mutans* adhesion to the nanostructured Ti surface (SEM micrographs).

**Figure 4 biomolecules-12-01515-f004:**
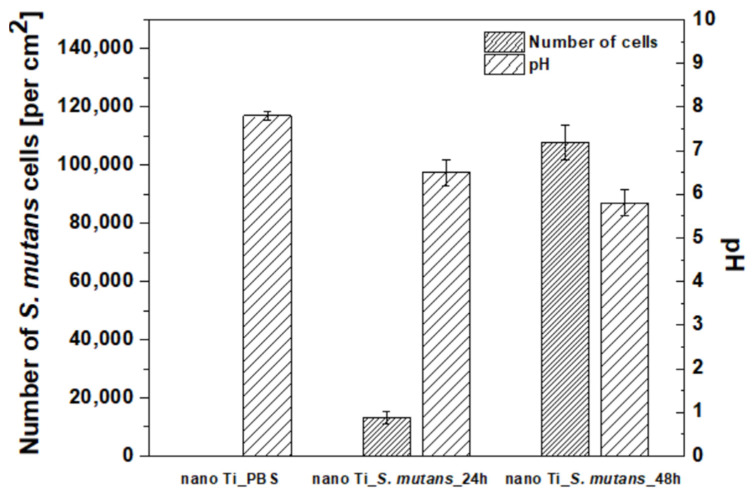
Calculated number of *S. mutans* cells per 1 cm^2^ of nanostructured Ti surface and pH value of the suspension measured after bacterial incubation.

**Figure 5 biomolecules-12-01515-f005:**
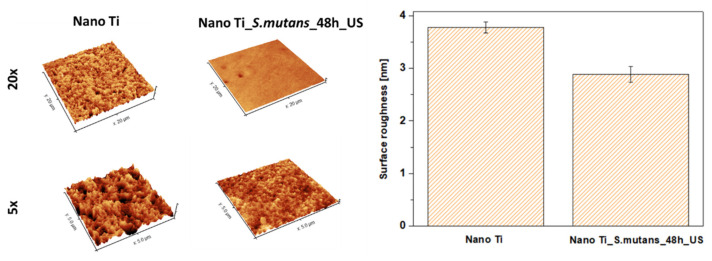
Surface roughness of nanostructured Ti in the polished state and after bacteria removal (AFM studies).

**Figure 6 biomolecules-12-01515-f006:**
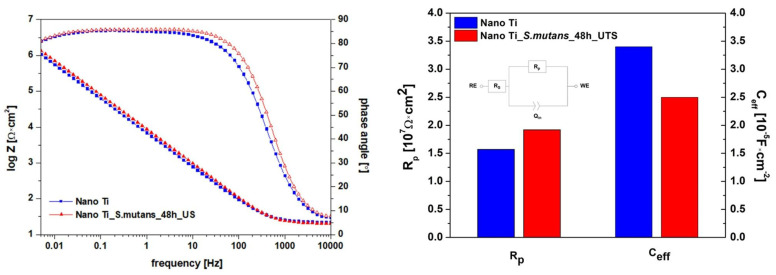
Corrosion resistance of nanostructured Ti in the polished state and after bacteria removal (electrochemical evaluation, EIS tests).

**Figure 7 biomolecules-12-01515-f007:**
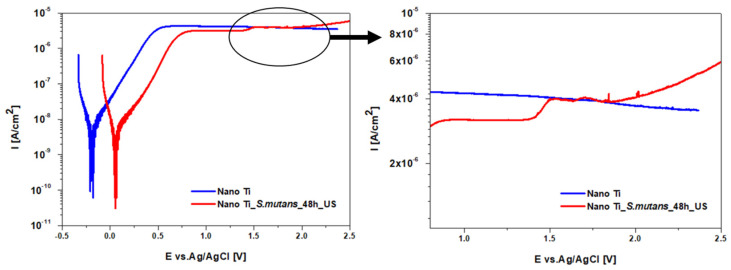
Corrosion resistance of nanostructured Ti in the polished state and after bacteria removal (electrochemical evaluation, PD tests).

**Table 1 biomolecules-12-01515-t001:** Description of tested samples.

Sample	Surface Preparation	Exposure	Surface Preparation after Exposure	Number of Samples
Nano Ti	Polishing	-	-	3
nano Ti_PBS (control)	Polishing	48 h of exposure in PBS	Sputtering with gold layer–SEM	1
nano Ti_*S. mutans*_24 h	Polishing	24 h of exposure in *S. mutans* suspension	Sputtering with gold layer–SEM	6
nano Ti_*S. mutans*_48 h	Polishing	48 h of exposure in *S. mutans* suspension	Sputtering with gold layer–SEM	3
nano Ti_*S. mutans*_48 h_US	Polishing	48 h of exposure in *S. mutans* suspension	Ultrasonication (bacteria removal)–AFM and corrosion	3

**Table 2 biomolecules-12-01515-t002:** Number of bacteria cells and pH value of the bacteria and control suspensions. Statistical significance of differences between values designated for different samples were confirmed by one-way ANOVA test (*p* < 0.05).

	Nano Ti_PBS (Control)	Nano Ti_ *S. mutans*_24 h	Nano Ti_ *S. mutans*_48 h
Number of *S. mutans* cells [per cm^2^]	0	13 176 ± 2 260	107 764 ± 5 808
pH	7.8 ± 0.1	6.5 ± 0.3	5.8 ± 0.3

**Table 3 biomolecules-12-01515-t003:** Values of arithmetic roughness (Ra) designated for the nano Ti in a polished state and after 48 h of incubation in *S. mutans* and subsequent bacteria removal. Statistical significance of difference between values designated for different samples was confirmed by one-way ANOVA test (*p* < 0.05).

	Nano Ti	Nano Ti_*S. mutans*_48 h_US
Ra [nm]	3.78 ± 0.10	2.89 ± 0.15

## Data Availability

Not applicable.

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
