# Peer review of "How Streptococcus mutans Affects the Surface Topography and Electrochemical Behavior of Nanostructured Bulk Ti"

_biomolecules, 2022, doi:10.3390/biom12101515_

Round 1

Reviewer 1 Report

For a long time I haven't review so well-written paper and such a well conducted study. The sentences are clear, the main topic has been researched and explained in an excellent way, well cited and supported with relevant results.

Only remark: figure 6 is missing (you have figure 5 and 7).

Author Response

We are grateful for giving us the opinion about our manuscript are for all of the kind words. We agree that Figure 6 is missing. We would like to confirm that in the revised version of the manuscript we corrected this mistake and changed the numeration of doubled Fig.5. Now the description of Fig.6. is as follows: Fig.6. Corrosion resistance of nanostructured Ti in the polished state and after bacteria removal (electrochemical evaluation – EIS tests).

Reviewer 2 Report

Revision for biomolecules

This work is good and well written, but it needs improvement to be published.

1-     Introduction -Page 2 line 11: describe “AFM” ( atomic force microscope) and all the other abbreviations, the first time you use them.

2-     Materials and methods -  page 2 line 91: were Ti disks sterilized before immersion in the solution with bactéria? Please describe how disks were manipulated until settlement.

3-     Line 139: highlight the groups tested and provide a table including the untreated control group and all the other groups tested, according to surface preparation and bacteria exposure,  describing how many times each experiment was replicated.

4-     Describe how data was collected, compared, and statistically analyzed, including p. values.

5-     Separate results in subtitles for each variable analyzed and provide a table of the results followed by graphics with standard deviation and p values.

6-     Discuss results in another topic right after.

Author Response

Reviewer 2 Comment:

This work is good and well written, but it needs improvement to be published.

Answer:

We are grateful for Reviewer opinion and for giving us the opportunity to submit the revised version of this manuscript. Below we included responses for particular Reviewer’s comments.

Comment 1:

Introduction -Page 2 line 11: describe “AFM” ( atomic force microscope) and all the other abbreviations, the first time you use them.

Response 1:

According to the Reviewer’s requirement, we described all of the abbreviations used for the first time in this study.

Comment 2:

Materials and methods -  page 2 line 91: were Ti disks sterilized before immersion in the solution with bactéria? Please describe how disks were manipulated until settlement.

Response 2:

Firstly, after polishing procedure, surfaces were evaluated at optical microscope with DIC contrast in order to verify if there are no any scratches or contaminations originated from polishing agent. Then, samples were cleaned in the 70% ethanol solution and dried before immersion in the solution with bacteria. As can be seen in Fig.3., no bacteria were observed for the control group (nano Ti discs incubated in PBS suspension for 48 h).

We would like to mention that before performing bacteria studies on titanium surfaces, the droplets with bacteria suspension were putted on the copper mesh and then characterized by SEM microscopy. This additional control allowed us to evaluate the size and morphology of S.mutans bacteria in our suspension. Finally we could compare the characteristics of bacteria applied on the cooper mesh with those finally cultured on the nanostructured Ti samples in order to be sure that we observed the growth of S.mutans.

Comment 3:

Line 139: highlight the groups tested and provide a table including the untreated control group and all the other groups tested, according to surface preparation and bacteria exposure,  describing how many times each experiment was replicated.

Response 3:

According to the Reviewer’s comment, in the revised version of the manuscript, we included table with all of the samples tested in our study. We would like to highlight that both bacteria exposure experiments (conducted for 24 h and 48 h) were performed for six independent samples. However, in case of 48 h of exposure, three of the six samples were ultrasonicated and devoted for further AFM and electrochemical tests. Selection of the samples after 48 h of exposure was justified by the slight decline of pH value, which could provide differences in the surface topography and thereby surface electrochemical response.

Table 1. Description of tested samples

Sample

Surface preparation

Exposure

Surface preparation after exposure

Number of samples

Nano Ti

Polishing

-

-

3

nano Ti_PBS (control)

Polishing

48 h of exposure in PBS

Sputtering with gold layer - SEM

1

nano Ti_S.mutans_24 h

Polishing

24 h of exposure in S.mutans suspension

Sputtering with gold layer - SEM

6

nano Ti_S.mutans_48 h

Polishing

48 h of exposure in S.mutans suspension

Sputtering with gold layer - SEM

3

nano Ti_S.mutans_48 h_US

Polishing

48 h of exposure in S.mutans suspension

Ultrasonication (bacteria removal) – AFM and corrosion

3

Comment 4:

Describe how data was collected, compared, and statistically analyzed, including p. values.

Response 4:

According to the Reviewer’s comment we provided statistical analysis using one-way analyses of variance (ANOVA) and post hoc Tukey test [1]. We evaluated statistical differences between (i) the number of bacteria per cm2 calculated after different times of bacteria exposure, (ii) pH values of the suspension evaluated after different time of bacteria exposure, (iii) surface topography (Ra and Rq parameters) evaluated for nano Ti in polished state and after bacteria exposure and further ultrasonication. As in literature data related to the metallic biomaterials [2, 3], we identified significant statistical difference for p <0.05.

Owing to providing statistical analysis we added following text to the Materials and Methods section:

“ Data related to the number of bacteria, pH and surface topography were statistically analyzed using one-way analysis of variance (ANOVA) and post-hoc Tukey tests. Significant statistical differences were identified for p<0.05”

Comment 5:

Separate results in subtitles for each variable analyzed and provide a table of the results followed by graphics with standard deviation and p values.

Response 5:

As was suggested by Reviewer’s we separated Results section and provided following parts:

  • Microstructure and surface topography of nanostructured Ti
  • Bacteria adhesion to the nanostructured Ti surface
  • Effect of bacteria exposure on the surface topography of nanostructured Ti
  • Effect of bacteria exposure on the electrochemical response of nanostructured Ti

Moreover, we provided additional tables with the numerical values of particular parameters presented in the graphs (precisely medium values, standard deviation values and information about the statistical significance of difference between the values of particular parameters obtained for different samples). Attached Tables are as follows:

Table 1. Number of bacteria cells and pH value of the bacteria and control suspensions. Statistical significance of differences between values designated for different samples were confirmed by one-way ANOVA test (p<0.05)

nano Ti_PBS (control)

nano Ti_

S.mutans_24 h

nano Ti_

S.mutans_48 h

Number of S.mutans cells [per cm2]

0

13 176 ± 2 260

107 764 ± 5 808

pH

7.8±0.1

6.5±0.3

5.8±0.3

Table 2. Values of arithmetic roughness (Ra) designated for the nano Ti in polished state and after 48h of incubation in S. mutans and subsequent bacteria removal. Statistical significance of difference between values designated for different samples was confirmed by one-way ANOVA test (p<0.05)

nano Ti

nano Ti_S.mutans_48 h_US

Ra [nm]

3.78±0.10

2.89±0.15

We would like to mention that in this study, for both states (nano Ti) and (nano Ti_S.mutans_48h_US), EIS corrosion studies were performed two times. Thereby, in the Fig.6. we presented average values of designated parameters. However, the main aim of this tests was to confirm the high protectiveness of the passive layer after bacteria incubation. This goal was accomplished and presented in the manuscript. Moreover, numerical values related to the corrosion resistance are included within the description of the results.

Comment 6:

Discuss results in another topic right after,

Response 6:

We would like to mention that all of the results are discussed in the Results and Discussion section. This section was improved at the revision stage by providing additional tables with numerical data, as was suggested by Reviewer. Considering the fact that such studies were performed for the first time for the promising, highly deformed titanium substrates, we decided to provide separate section for the Future Directions of studies.

  1. Yu F, Addison O, Davenport AJ (2015) A synergistic effect of albumin and H2O2 accelerates corrosion of Ti6Al4V. Acta Biomater 26:355–365. https://doi.org/10.1016/j.actbio.2015.07.046
  2. Mace A, Khullar P, Bouknight C, Gilbert JL (2022) Corrosion properties of low carbon CoCrMo and additively manufactured CoCr alloys for dental applications. Dent Mater 38:1184–1193
  3. Hedberg YS, Žnidaršič M, Herting G, et al (2019) Mechanistic insight on the combined effect of albumin and hydrogen peroxide on surface oxide composition and extent of metal release from Ti6Al4V. J Biomed Mater Res Part B Appl Biomater 107:858–867

Reviewer 3 Report

I thought that the author should be mentioned structural analysis (STEM analysis) more in detail. May be EBSD analysis would be suitable for the nanostructured Ti .

Author Response

Reviewer 3 Comment

I thought that the author should be mentioned structural analysis (STEM analysis) more in detail. May be EBSD analysis would be suitable for the nanostructured Ti .

Answer:

We are grateful for given remark. According to the Reviewer’s suggestion, we provided the description of STEM analysis within the manuscript (Fig.1 shows the microscopic evaluation of the microstructure and surface topography of nanostructured Ti fabricated by large plastic deformation. It can be noticed that multiple-pass HE process resulted in the formation of well-defined nanometric grains with the high dislocation density in their interiors. – page 4 in the revised version of this manuscript). We have to agree that detailed characterization of the nanostructure would be valuable for this study. As was suggested by Reviewer, we considered EBSD to gain knowledge related to the grain size and grain boundaries misorientation. However, it have to be mentioned that the material tested in this study is composed of nanometric grains with high dislocation density in their interiors. According to literature, the application of EBSD technique to routine characterization of homogenous nanostructured materials is not possible owing to its limited spatial resolution [1]. Our trials to characterize nanostructured Ti by EBSD were also not successful, precisely, we obtained very limited number of indexed grain boundaries which makes this characteristic unreliable.

  1. Trimby PW (2012) Orientation mapping of nanostructured materials using transmission Kikuchi diffraction in the scanning electron microscope. Ultramicroscopy 120:16–24

Round 2

Reviewer 3 Report

I don't have any questions.